# Examination of Local Plan Changes from a Value Capture Perspective: Istanbul Case

**Numan Kılınc * and Sevkiye Sence Turk**

Urban and Regional Planning, Istanbul Technical University, İstanbul 34467, Turkey; turkss@itu.edu.tr
* Correspondence: kilincnum@itu.edu.tr

**Abstract:** Local governments have an increasing tendency to capture the value increase occurring as a result of partial interventions into local plans. The basic acceptance behind this is that value definitely will increase as a result of partial interventions. However, all partial interventions always cannot lead to an increase in value. There can be also partial interventions in which the value does not change or even decreases. The aim of this study is to identify the value capture capacity of local plan changes as partial interventions, and to discuss this capacity in terms of the balance between betterment and compensation. Istanbul, which is one of the cities where the effects of neo-liberal policies are most intense and where local plan changes are common, was chosen as the study area. In the first stage of the study, the spatial distributions of 17,369 plan changes approved by the Istanbul Metropolitan Municipality Council between 2009–2018 were examined. In the second stage, the value capture capacities of the plan changes grouped by subject, were determined by interviewing 46 people working in different areas of the planning discipline. The findings of the study demonstrate that although the plan changes are spread throughout the metropolitan area, they are concentrated in the central and secondary central districts where the accessibility value is high. The interviewees emphasized that the plan changes made as a result of private-sector demand and the plan change for the improvement of the infrastructure increase the value of the land and that the plan changes within this scope have value capture capacities. On the other hand, according to the findings of the study, some plan changes reduce the value of the land because of restricting the property rights on the land. Plan changes in this group are needed to be compensated fairly and equitably. Thus, the balance between betterment and compensation would be achieved.

**Keywords:** plan changes; value capture; land value; Turkey; Istanbul

## 1. Introduction

Public land value capture refers to the capture of part or all of the value increase occurring in land as a result of the efforts of public bodies rather than from those of private landowners, to the public sector for the purpose of financing public activities [1–6]. Public land value capture has recently become a subject that has recieved increasing attention [7–12]. A reason for this is that real estate is an attractive target for tax authorities [7–13]. Another reason is that value capture creates a significant resource for local governments to provide public services [6–8,10,11,14]. On the other hand, the new urban agenda emphasises that the income obtained through public value capture should not be considered a source of income, but as a tool to encourage the desired urban development, to exercise a controlling effect on land, and to be used for new investments [15–18]. Similarly, the World Bank sources, too, recommend that value capture tools should be used in urban areas to lighten the burden of the public [19,20].

The idea that the value of the land was created by the state and, therefore, should be collected for the public, is not new. In his book *Progress and Poverty*, the American philosopher Henry George suggested the idea of a 'single tax' in 1879, and maintained that this idea would meet all public needs of the community in the case of a continuous

flow of payment from land to the state administration [21]. The idea to use the value increase stemming from public activities for public purposes, pioneered by Henry George, was adopted and applied by many countries [2,3,5]. For instance, in the British planning system, the correlation between state regulations and property values goes back to the late nineteenth century and can be seen in legislative provisions in the form of 'betterment and compensation'. Those provisions pioneer the matter of returning the increasing property values to the public [22]. Based on that thought, the British Town and Country Planning Act of 1947 reshaped the entire system, and, by introducing the idea of the 'development rights', put in place a system under which each development is approved by 'planning permission' on a 'case-by-case basis' [14,23].

Different aspects of value capture have been discussed in the international literature. The most important among these are the rationale underlying value capture [3,7,8,12,24,25]; the effectiveness of value capture tools [4,10,26–31]; how much of the value may be captured [11,14,32–35]; the sharing of the captured value among actors [30,36], and the institutionalisation of value capture in planning law [1,2,11]. Although value capture tools vary by country, they are generally divided into three basic classes: macro, direct, and indirect [3]. Among these tools, while macro value capture focuses on more comprehensive planning activities, direct value capture mostly focuses on the capture of the value increase occurring in land as a result of interventions performed by a public authority apart from the property owner's acts, to the public. Indirect value capture, however, is mostly used for internalising the negative externalities of investments on a local scale [11]. However, criticisms have been put forward to the effect that, in general, value capture tools only focus on achieving the value increase occurring as a result of public activities, and that value losses resulting from public activities are mostly ignored, which gives rise to an inability to strike a balance between betterment and compensation [37].

This is mostly seen in value capture approaches aimed at partial planning interventions. An increase in the development rights in the area through partial interventions, and the value of land also increases as a natural consequence thereof. This value increase in land is also recognised and supported by local governments as an increase in the tax imposed on property [12,13,31,38]. Or an accumulation of capital is ensured in local governments to be used in gapping the negative externalities and the shortcomings in social and technical infrastructure stemming from such interventions, through the obligations imposed on investors in consideration of the addition of private sector investments to an urban venue [7,10,11,35,39]. In this case, local governments obtain income both directly and indirectly [31,40]. However, depreciation is also possible as a result of partial interventions. However, this is mostly ignored.

There are studies in the literature focused on capturing value increase stemming from partial interventions [1,6,8]. These studies concur that value definitely increases as a result of interventions. Nonetheless, not all partial interventions may cause an increase in value. There may be interventions where value remains unchanged, or even decreases [10,35]. Nevertheless, the legal framework of many countries is shaped on the assumption that partial interventions mostly increase land values. On the other hand, regulations addressing cases where partial interventions decrease or depreciation [2,37,40–44] are inadequate. Cases where value decreases or does not change as a result of partial interventions, and the balance between betterment and compensation, have not been adequately discussed in the literature. The objective of this study is to examine the value capture capacity of local plan changes as partial interventions, and to discuss this capacity in terms of the balance between betterment and compensation.

In Turkey, since the 1980s, governments have targeted policies that focused on development, privatising public resources, and supporting the construction sector, in order to attract private sector investments, mostly by ignoring social and environmental negative impacts [45]. After 2002, in particular, the neo-liberal urbanisation tendency has increased even further in Turkey due to low interest rates and changes in legislation, causing the housing and construction sector to grow rapidly [46]. In Turkey, Istanbul is the city where

the impact of neo-liberal economic policies are felt most intensely [47]. Many plan changes have been made in Istanbul as a result of the adoption of a market-oriented planning understanding [46,48,49]. As many as 17,369 plan changes approved by the Istanbul Metropolitan Municipality Council alone were made between 2008 and 2018 [50]. Since Istanbul has a dynamic structure in terms of plan changes and investments, examining its capacity of value capture caused by plan changes in terms of the balance between betterment and compensation can make a contribution to cities of other countries facing neo-liberal pressures.

In the study, 17,369 local plan changes approved by the Istanbul Metropolitan Municipality Council between 2009 and 2018 were grouped by subject, and the value capture capacities were identified by interviewing 46 individuals working in different fields of the planning discipline. In the study, how a balance between "betterment" and "compensation" may be achieved in the return of the value to the public is also discussed.

The article consists of five chapters. Value capture is discussed in the second chapter following the introduction. The third chapter gives information on value capture policies and plan changes in the Turkish planning system. In the fourth chapter, the plan changes in the Istanbul Metropolitan Area, which was chosen as the study area, are examined in detail within the scope of value capture. The fifth chapter contains the conclusion and a general evaluation.

## 2. Value Capturing Policy and the Betterment—Compensation Balance

Public value capture refers to capture all or part of the value increase, which occurs in land as a result of interventions made by a public authority, apart from the actions of property owners, to the public sector for the purpose of financing public activities [1–3,6]. While this value increase may be achieved indirectly by capturing it into public revenue through the use of taxes, charges, or other financial tools, it may also be achieved directly through on-site betterments made for the purpose of providing benefits to a larger community [3].

Alterman [11], suggests that the idea developed by Britain, which envisages that increased value should bring certain monetary obligations to landowners, also gives rise to the question of whether or not a right to compensation would be appropriate in the opposite case, that is, in the case of a value decrease as a result of the state's activities, and by whom this compensation should be paid. She categorises the tools used in ensuring the public return of the value increase resulting from public intervention, into three main classes: macro, direct and indirect tools. She classifies the nationalisation, expropriation, land banking, and land readjustment practices as macro value capture tools. Direct value capture, however, is defined as the act of capturing the value increase occurring in land as a result of a public activity, back to the public, mostly through a tax practice, while indirect value capture has a motivating logic for meeting the costs of public infrastructure and facilities needed to support newly-developing areas [11]. In many countries, local governments adopt indirect value capture tools rather than direct value capture tools [4,51]. The reason is that the impact of indirect value capture tools is felt less by citizens compared to that of direct value capture tools, and that direct value capture tools are supported by local governments as they are capable of ensuring direct income for urban infrastructure [3,10].

On the other side are voters' unwillingness to pay higher taxes, the fact that many services have high costs, and voters' expectations towards developed services [52]. Therefore, local governments are increasingly turning to the use of financial tools that are less conspicuous to voters compared to direct taxes or charges [3,6,7]. There is also an idea in the literature that maintains that planning activities that are public acts do not always increase land value, but actually decrease it, which should be compensated by the public [2,11,37,40]. The idea of compensating the value decrease stemming from planning activities was first suggested by Hagman and Misczynski (1978). The idea that compensation should be provided in the case of value decreases resulting from planning decisions, suggested by Hagman and Misczynski, was based on fairness. While plan decisions increase value in some areas [4,11,53], they restrict landowners' rights of use in some other areas. In such

cases, the value of the land shall also drop in connection with the decision on how the land should be used. Accordingly, some plan decisions, while benefitting the community, cause property owners to suffer difficulties [2,40,44]. In this sense, the compensation of the decreasing land value reduces the potential of the planning activities to become the subject of politics, causing public acts to become more equitable [37,40].

The right to compensation refers to the amount that is necessary to be paid by planning authorities to landowners in connection with the decreases in land value stemming from decisions on the use of land [3]. Therefore, the compensation is related to the compensation for plan decisions restricting use rather than the expropriation price of the land allocated to public use. The amount of compensation generally corresponds to the difference between the value of the land before and after the plan decision [2,40]. However, since land value increases differently in different stages of the planning process [54], the land's value must be considered in conjunction with the different stages of the planning when assessing the scope of the compensation rights [2].

Various countries have developed different methods to compensate value decreases. Alterman [40], refers to Canada, Britain, Australia, France and Greece, where regulatory framework relating to the compensation of value decrease resulting from planning is very limited, as countries that have adopted a minimum compensation right policy. She indicates Finland, Austria and the United States as countries that have regulatory frameworks that grant a somewhat broader range of compensation rights although containing ambiguities. On the other hand, she classifies Poland, Sweden, Israel and the Netherlands, which offer medium level but uncertain compensation rights and have regulatory frameworks for compensating even the partial restrictions in keeping with laws and court decisions, as countries granting a broad right of compensation [40]. In Poland, which offers broad compensation rights, provisions about the compensation of value decrease first came into force through the Law of 1994, and was re-regulated through the Spatial Planning and Development Act that was promulgated in 2003. While this law offers a choice between expropriation of land by the municipality and the compensation of the actual loss, in the event that the land has been restricted to a large extent by a plan decision, it offers the choice of the loss being compensated by the municipality in partial restrictions. However, it cannot be used most effectively as a very small number of landowners are aware of this right [55]. Havel [2], on the other hand, emphasises that many local governments do not have the financial strength to pay the compensation relating to value capture, in which case the planning officials are faced with a dilemma of preparing the plans and facing the financial burdens or giving up control over urban development.

### 3. Value Capture Policy on Plan Changes in Turkey

In Turkey, there are various value capture tools for grabbing the increases in land value resulting from public activities. They may be classified as macro, direct and indirect tools. The macro value capture tools may be summarised as (1) nationalisation, (2) expropriation, (3) land readjustment, (4) voluntary method; direct value capture tools as (5) participation share, (6) Article 80 of Income Tax Law No. 193 (value increase tax), (7) property tax, (8) share of the value increasing as a result of plan changes; and indirect value capture tools as (9) transfer of development rights, (10) developer's obligations, and (11) plan notes [28,30,51]. Since this study focuses on value capture relating to plan changes, the other value capture tools are left outside its scope.

In urban planning, plan changes and infrastructure investments are among the primary causes of the increase in the value of land. The higher the development rights resulting from plan changes, the higher the increase in the value of the land. However, in Turkey, no rule was in place for ensuring the return of the increasing value in connection with plan changes back to the public until the supplementary amendment to Reconstruction Law No. 3194, that came into force on 14 February 2020, and the "Regulation on Value Increase Share Relating to Development Plan Changes", that came into force on 15 September 2020. Therefore, this increased the pressure for change on the local plans. The value increase that

occurred in plan changes carried out by the private sector to maximise profit was shared between the investor and the landowners, and the return of this increase to the public could not be achieved either. Due to political concerns, the issue of the return of the income caused by plan changes back to the public were mostly ignored, and allowed the facilitation of the increase in investments. However, many changes made in the plans in order to integrate the investment decisions into the urban space have caused some problems in the cities.

Thus, the cities started to develop not as prescribed by the plans, but through plan changes. Furthermore, plan changes performed due to requests made by landowners and investors are not concerned with public benefit, and cause certain problems that are difficult to control, such as a decrease in social and technical infrastructure areas in urban planning, inadequate transport and infrastructure, and an increase in population and building density above that which was planned [18,50,56–60].

Due to the new legal framework in 2020, plan changes can only be made in public parcels and plan changes made on a parcel basis because plan changes to be made on a single-parcel basis were prohibited by the Reconstruction Law in 2020. According to the new legal framework, value increase payments would be made by landowners no later than upon the first sale of the real estate or when getting a construction permit. However, this legal framework, too, assumes that plan changes cause value increase. There is no consideration given against the possibility of value decreases resulting from plan changes. On the other hand, the purpose of this framework was to capture the increasing value to the public. This may be considered a direct value capture in terms of the purpose of the regulatory framework, and when considering that it was handled through a regulatory act across all of Turkey. However, from the aspect of the collection method under this regulatory framework, it also features indirect value capture characteristics due to the fact that decisions are made on a case-by-case basis and that the collections are made directly from developers and landowners. It also resembles direct value capture tools in terms of how the collected money is used. In this case, the legal framework has regulated a hybrid nature that includes both direct and indirect logic. Although the existing regulatory framework specifies how this amount will be collected and how and among which agencies it shall be shared, the local services in which this amount will be used or in which city/city population it will be used for are not clear in the new framework.

## 4. Analysis of Local Plan Changes from a Value Capture Perspective: Istanbul Case

### 4.1. Study Area

Istanbul is located strategically, connecting Europe and Asia, and this settlement with a rich history covers an area of 5400 km$^2$ [61]. Since the 1980s, likewise other global cities [62], the production, commerce and consumption systems in Istanbul have become increasingly globalized, and the spatial, social and economic structure in Istanbul changed dramatically [63]. Therefore, especially the transportation infrastructure was improved, and the pressure to develop prestigious, large-scale commercial and residential projects emerged [48,49]. For large-scale developments, being close to locations where activities are bustling or having a central location [64] provides a great advantage in terms of accessibility. Investments, which not only play a role in the economic activities in global cities but also affect the physical construct of cities, gravitate toward high-profile investments including transportation infrastructure (such as important transportation nodes, transportation axis, etc.), hospitals, universities, parks, and international conference centres [65], thus implying that Istanbul should allow for large-scale infrastructure projects to attract global investments.

After 1980, the impacts of the changes in construction rights and in land uses under the influence of globalization were observed in Istanbul. Moreover, such changes became apparent, particularly in the service sector [66], with the increase in the flow of foreign investment, the foreign migration Istanbul received between 1990 and 2000, and the sectoral mobility due to the increase in population [67]. After the 1990s, large-scale

housing investments enabled by the private sector or by public-private partnerships also proliferated [68]. Starting with the 2000s, Istanbul entered a restructuring process under the influence of neo-liberal urban policies [69–71]. This restructuring process took shape through an explosion in the volume of construction, which was dependent on the real estate market [46,72]. The rapid increase in the real estate market in Istanbul after the 2000s brought along a reorganization process in the city [46,69,70]. Thus, with the transformation of urban land uses in Istanbul due to globalization and neo-liberal policies, the amount of office spaces multiplied on the European side, while secondary offices and large-scale housing investments became prevalent on the Asian side [71]. Large investments played a significant role in changing the urban land use density and identifying the direction of urban development [72]. During this period, a significant increase was observed, especially in the number of mega projects [73–75] and large-scale housing projects [76,77]. This time, with the impact of the increasing population and capital, the urban spaces started facing the pressure of construction and functional change. Therefore, the number of requests for plan changes increased.

The fact that the city centre was inclined to decentralize [67] led to a multi-centre structure as the service and commercial land uses differed from each other in terms of their spatial behaviours [78]. Until the 1960s, Istanbul developed as a single-nucleus city, and when the Bosporus Bridge and connecting highways began operating in 1973, the layout of the city evolved into a multi-centre structure. The Bosporus, which also constitutes a natural edge, triggered the transition to a multi-centre layout so as to enable mobility on both sides of the city. The emergence of the multi-centre layout is also an outcome of the improved transportation network to enable access to the airport and to the international transportation system; and thus, numerous centres emerged along these transportation routes [66]. With reference to the district hierarchy suggested by Berkoz and Eyuboglu (2007), by the Istanbul Metropolitan Municipality (IMM, 2009) and by Turk (2017), the multi-centre layout in Istanbul is presented in Figure 1. The hierarchy consists of: (i) central districts, (ii) sub-central districts, and (iii) peripheral districts (Figure 1).

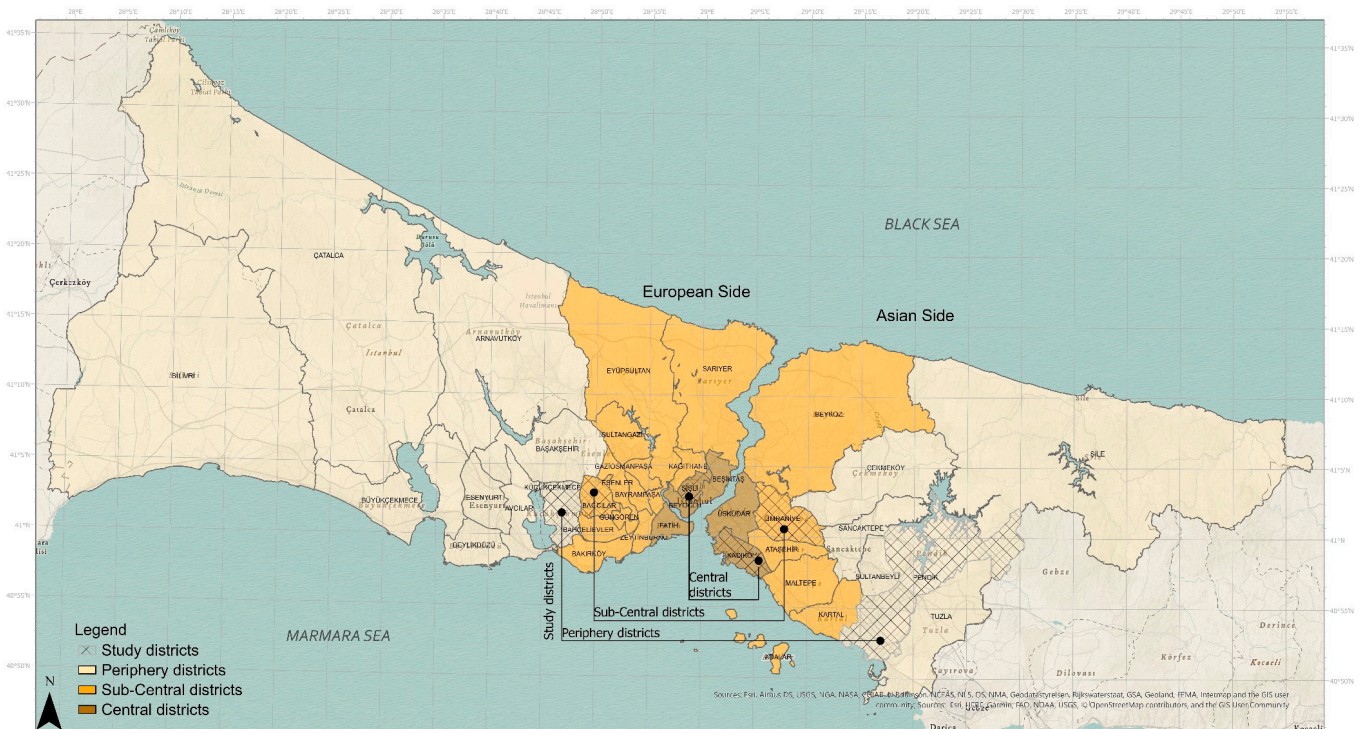

**Figure 1.** Hierarchy of districts in Istanbul (Kılınc and Turk, 2021).

*4.2. Methodology*

The methodology of the study consists of two stages. In the first stage, the municipal council resolutions relating to all plan changes approved by the Istanbul Metropolitan Municipality Council between 2009 and 2018 were examined. The data obtained were organised and transferred to the digital environment. A total of 17,369 plan changes made between 2009 and 2018 were classified according to their contents. Twelve sub-categories were obtained as a result of this categorisation. However, the last three categories were not included in the second stage of the analysis as they were changes not defined legally by the plan change procedure (Types 10–12). Therefore, while the evaluations in the first stage of the study were performed on 17,369 plan changes, the evaluations in the second stage were performed over 15,397 plan changes that remained after the mentioned types were eliminated.

In the second stage, the plan changes classified by content were discussed with forty-six individuals working in various areas of the planning discipline. The value capture capacities relating to these plan change types were identified. For this purpose, the interviewees were shown 1–2 real examples that best reflected the nine groups emerging after the classification made for the plan changes. They were provided with summary information and figures relating to these samples. These plan changes chosen best reflect each sub-group. They had not been cancelled by court order before, and they have implemented.

The 46 interviewees working in different areas of the planning discipline were interviewed within the scope of the study. The interviewees were asked to choose one of the three options: "value-increasing (+1)", "value-decreasing (−1)" or "ineffective (0)" in connection with the plan changes offered from 9 sub-headings as samples. Also, they were asked to evaluate the value capture capacities of the classified plan changes, as "no capacity (0)", "very low capacity (1)", "low capacity (2)", "medium capacity (3)", "high capacity (4)" and "very high capacity (5)".

The professional experiences of the specialists working in different fields of the planning discipline may have an effect on their approaches and evaluations relating to plan changes. Therefore, the interviewees were chosen from among individuals who are active in different fields of the planning. Furthermore, since the interviewees' experiences could influence their evaluations, they were chosen from among individuals with a minimum of ten years' uninterrupted experience. The interviewees' positions may also have an influence on their evaluations. For that reason, interviewees from among different position groups among academics, private sector employees and public sector employees were taken into consideration. Sixteen of the interviewees were academics: five were professors, six were associate professors, and five held PhD degrees. The position variable was also taken into consideration in the sample of private sector planners. Fifteen interviewees were chosen from among planners who had been employed by the private sector for at least ten years. Six of these were owners of private firms while nine were employees of private firms. The position variable was also considered in the sample consisting of public sector planners so that public sector officials from different positions were selected. Fifteen of the interviewees were planners who had been employed in the public sector (metropolitan municipality, district municipality, Provincial Directorate of the Ministry of Environment and Urbanisation, and other public agencies) for a minimum of ten years. Details relating to the interviews made within the scope of the analysis are provided in Table 1.

**Table 1.** Details of the interviews conducted within the scope of the research.

| Interview No/Date | Academician | Interview No/Date | Public Sector | Interview No/Date | Private Sector |
|---|---|---|---|---|---|
| Interview-1 11 March 2019 | Academician, Professor Istanbul Technical University | Interview-17 25 March 2019 | Ministry of Environment and Urbanization (Director) | Interview-32 22 March 2019 | Owner of Planning Office |

**Table 1.** *Cont.*

| Interview No/Date | Academician | Interview No/Date | Public Sector | Interview No/Date | Private Sector |
|---|---|---|---|---|---|
| Interview-2 12 March 2019 | Academician, Professor Istanbul Technical University | Interview-18 25 March 2019 | Ministry of Environment and Urbanization | Interview-33 24 March 2019 | Owner of Planning Office |
| Interview-3 18 March 2019 | Academician, Professor Istanbul University | Interview-19 27 March 2019 | Istanbul Metropolitan Municipality (Director) | Interview-34 24 March 2019 | Owner of Planning Office |
| Interview-4 4 April 2019 | Academician, Professor Istanbul University | Interview-20 27 March 2019 | Istanbul Metropolitan Municipality (Chief) | Interview-35 25 March 2019 | Owner of Planning Office |
| Interview-5 8 April 2019 | Academician, Professor Sakarya University | Interview-21 27 March 2019 | Istanbul Metropolitan Municipality | Interview-36 2 April 2019 | Owner of Planning Office |
| Interview-6 11 March 2019 | Academician, Assoc. Professor Istanbul Technical University | Interview-22 12 March 2019 | Fatih Municipality (Director) | Interview-37 3 April 2019 | Owner of Planning Office |
| Interview-7 13 March 2019 | Academician, Assoc. Professor Istanbul University | Interview-23 14 March 2019 | Pendik Municipality (Chief) | Interview-38 22 March 2019 | Employed in Planning Office |
| Interview-8 15 March 2019 | Academician, Assoc. Professor Yıldız Technical University | Interview-24 14 March 2019 | Üsküdar Municipality (Director) | Interview-39 22 March 2019 | Employed in Planning Office |
| Interview-9 18 March 2019 | Academician, Assoc. Professor Istanbul University | Interview-25 15 March 2019 | Kadıköy Municipality (Director) | Interview-40 22 March 2019 | Employed in Planning Office |
| Interview-10 20 March 2019 | Academician, Assoc. Professor Marmara University | Interview-26 19 March 2019 | Istanbul Registry Office | Interview-41 24 March 2019 | Employed in Planning Office |
| Interview-11 21 March 2019 | Academician, Assoc. Professor Fatih Sultan Mehmet University | Interview-27 13 March 2019 | Ministry of Culture and Tourism (Director) | Interview-42 24 March 2019 | Employed in Planning Office |
| Interview-12 19 March 2019 | Academician, Assist. Professor Gebze Technical University | Interview-28 13 March 2019 | Ministry of Culture and Tourism | Interview-43 25 March 2019 | Employed in Planning Office |
| Interview-13 13 March 2019 | Academician, Assist. Professor Yıldız Technical University | Interview-29 2 April 2019 | TOKİ (Mass Housing Autority) (Chief) | Interview-44 25 March 2019 | Employed in Planning Office |
| Interview-14 14 March 2019 | Academician, Assist. Professor Konya Technical University | Interview-30 2 April 2019 | TOKİ (Mass Housing Autority) | Interview-45 2 April 2019 | Employed in Planning Office |
| Interview-15 18 March 2019 | Academician, Assist. Professor Istanbul University | Interview-31 26 March 2019 | Ministry of Transport and Infrastructure (Director) | Interview-46 3 April 2019 | Employed in Planning Office |
| Interview-16 18 March 2019 | Academician, Assist. Professor Medeniyet University | | | | |

### 4.3. Findings

#### 4.3.1. First Stage: Classification of the Plan Changes

According to the findings relating to the first stage of the study, a total of 17,369 plan changes were made in Istanbul by the private and public sectors between 2009 and 2018. While 10,152 (58.4%) of the 17,369 plan changes made across Istanbul were changes requested by the private sector (property owners, investors), 7217 (41.6%) were those requested by the public sector (district municipality, metropolitan municipality, other public offices and agencies). The proposals for changes filed by landowners or investors were considered as private sector requests, and those made by district municipalities, the metropolitan municipality or other public offices and agencies were considered as public sector requests. The plan changes made across Istanbul were mostly performed in keeping with requests by the private sector indicates that the city was shaped in line with the requests of the private sector rather than the plan targets. 5994 of these changes were made in districts located on the Asian side while 11,375 were made in those on the European side. This demonstrates that plan changes mostly occurred on the European side. It may be thought that this is caused by the fact that the European side contains the primary centre of the city as well as many secondary centres. Furthermore, of the 17,369 plan changes made across Istanbul, 2667 were made in the central districts, 8790 in secondary central districts, and 5912 in peripheral districts (Figure 2). Plan changes concentrate on these districts especially since accessibility and land prices are high in the primary central and secondary central districts. Figure 3 shows the distribution of plan changes made on a parcel basis.

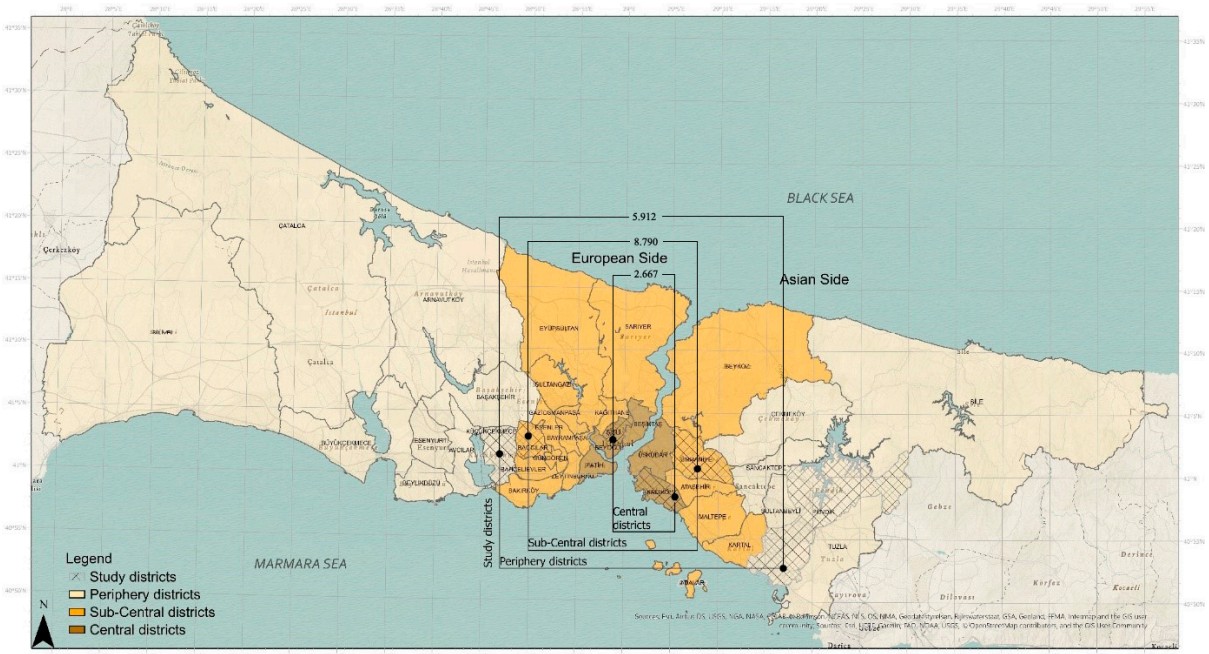

**Figure 2.** Number of plan changes in district levels.

Within the scope of the study, 12 groups emerged as a result of the classification of the plan changes (Table 2).

However, the three change groups (Types 10, 11, 12) were not included in the analysis in the second stage as they did not conform to the plan changes defined as legal. Therefore, 9 groups are taken into consideration in the classification of the plan changes (Table 2).

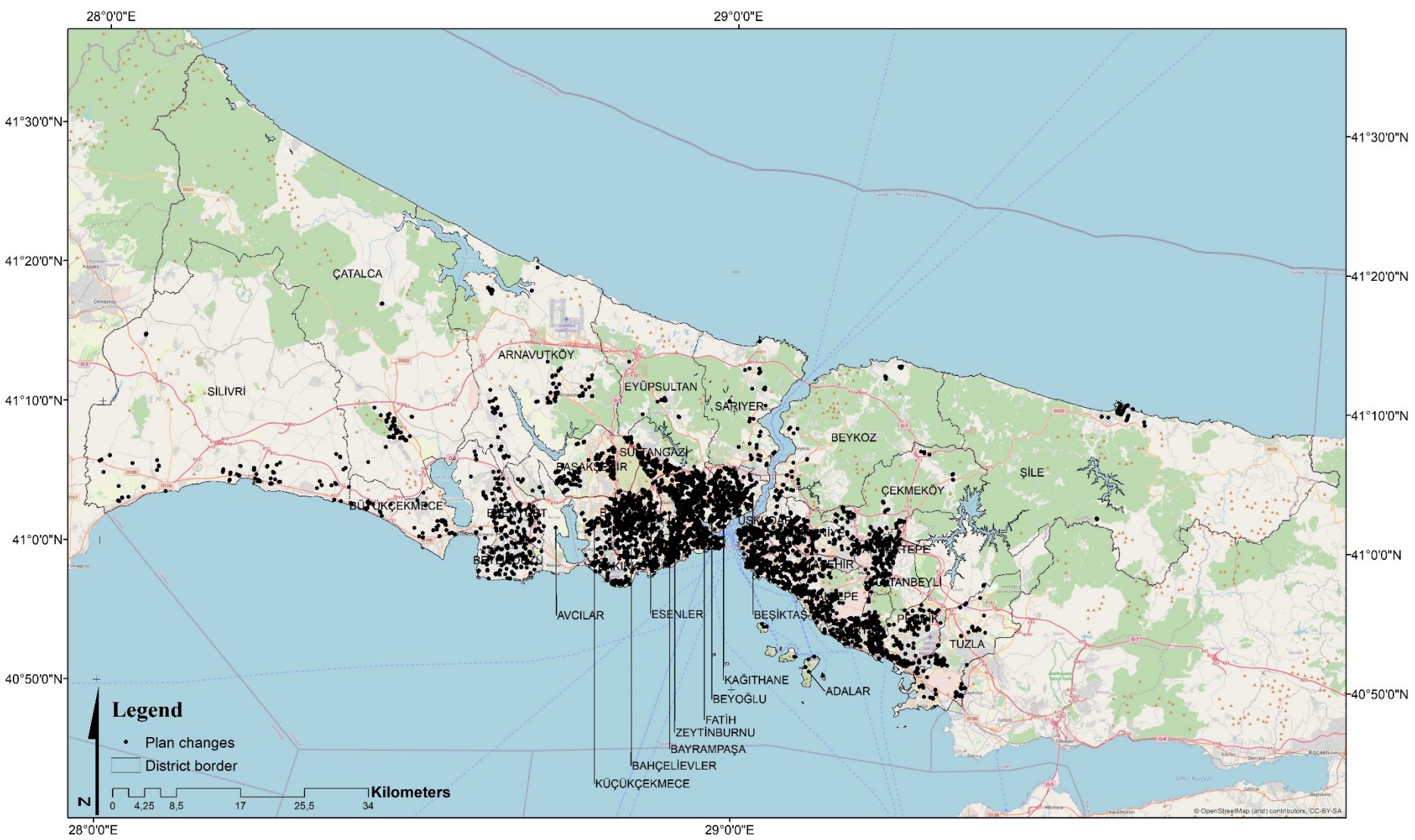

**Figure 3.** Distribution of plan changes made in Istanbul between 2009–2018 at the parcel scale (produced by the authors, 2021).

**Table 2.** Classification of plan changes between 2009–2018 in Istanbul.

| Groups | Scope of Plan Changes | Total |
|---|---|---|
| TYPE 1 | Changes from a Social and Technical Infrastructure Area to Another Social and Technical Infrastructure Area | 2126 (12.2%) |
| TYPE 2 | Changes from a Social and Technical Infrastructure Area to a Usage Area | 4031 (23.2%) |
| TYPE 3 | Changes from a Usage Area to a Social and Technical Infrastructure Area | 2600 (15%) |
| TYPE 4 | Changes Relating to Transport | 985 (5.6%) |
| TYPE 5 | Changes Relating to an Increase in the Floor Area Ratio (FAR) | 2877 (16.6%) |
| TYPE 6 | Changes Relating to a Decrease in the Floor Area Ratio (FAR) | 57 (0.3%) |
| TYPE 7 | Changes Relating to Usage Area' (Causing Density, Transport and Environmental Burden) | 1714 (9.9%) |
| TYPE 8 | Changes Relating to Usage Area' (Decreasing Density, Transport and Environmental Burden) | 154 (0.8%) |
| TYPE 9 | Changes Relating to Private Social Technical Infrastructure Area | 853 (5.0%) |
| | Total that taken in the analysis | 15,397 (88.6%) |
| TYPE 10 | Changes Relating to Plan Implementation Practices | 866 (5.0%) |
| TYPE 11 | Error Correction | 429 (2.5%) |
| TYPE 12 | Off-topic Changes | 677 (3.9%) |
| | General Total | 17,369 (100%) |

Within these groups, Types 2, 5, 7 and 9 are seen to be effective in the Istanbul Metropolitan Area. The plan changes in this scope cover changes emerging as a result of market dynamics rather than public needs. Such change requests are made by developers and landowners. They are changes that aim at acquiring greater development rights. For this reason, they involve private benefits. The outcomes of these three types of changes are also important. The amount of social and technical infrastructure decreases through Type 2 plan changes. The reason is that, through these changes, the existing social and technical infrastructure areas are opened to construction. In most cases, it is also possible that the removed social and technical infrastructure areas may not be replaced. In Istanbul, green area standards are considerably low among social and technical infrastructure areas [79]. These changes, besides having a direct impact on urban quality, also disrupt the balance between public and private spaces due to the decrease in public outdoor spaces. Type 5 plan changes, on the other hand, cause an increase in construction and population density. It is necessary that the city's transport system and social and technical infrastructure also grow accordingly. However, in spite of the density increasing in practice, no growth may be obtained in transport and in the social and technical infrastructure. The function of the area is changed through Type 7 plan changes, but this function change involves certain unfavourable aspects, such as an increase in transport activities and the inadequacy of infrastructure areas. Through Type 9 plan changes, urban use areas or facility areas set aside for public use are included into private facility areas, such as private education and private healthcare. Plan changes within the scope of Types 2, 5, 7 and 9 are among the

tools used in including the private sector investments, carried out in line with the pressures of neo-liberal economic policies, into the plans. In Figures 4 and 5, sketches of the plan changes are shown.

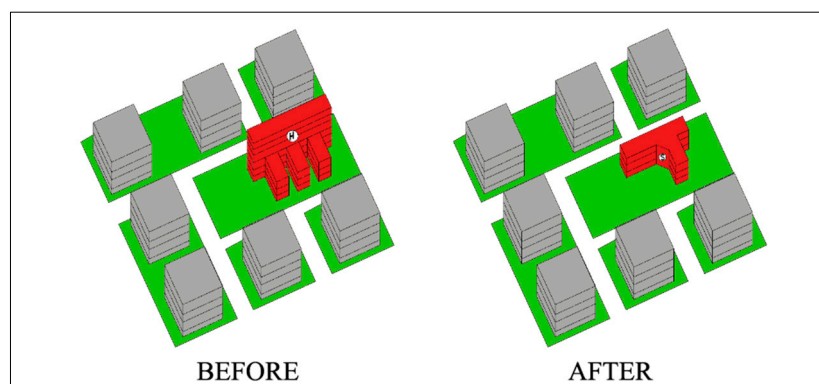

**Type 1.** Changes from a social and technical infrastructure area to another social and technical infrastructure area

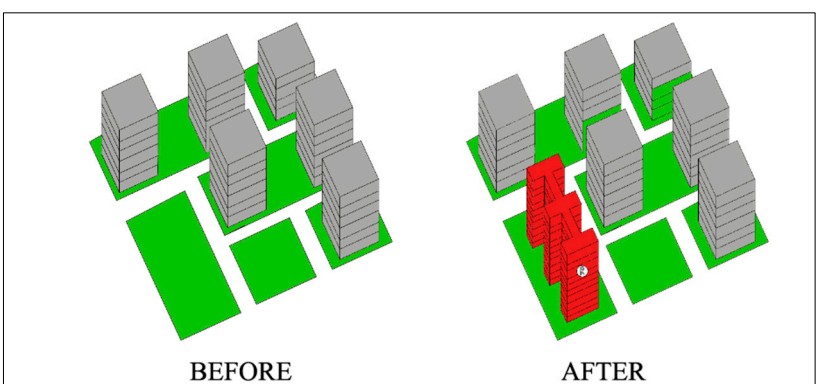

**Type 2.** Changes from a social and technical infrastructure area to a usage area

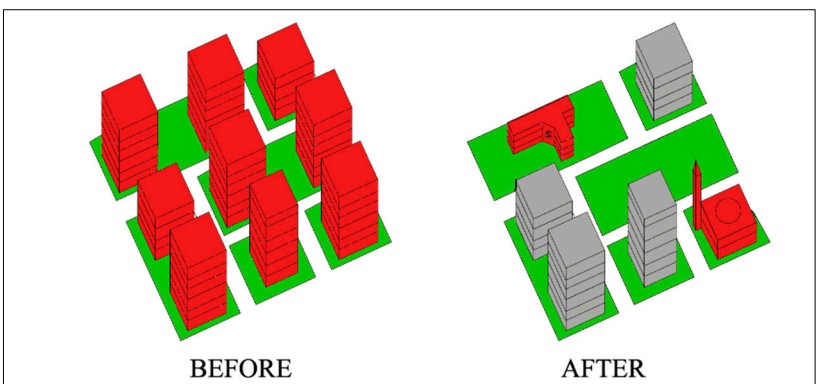

**Type 3.** Changes from a usage area to a social and technical infrastructure area

**Figure 4.** *Cont.*

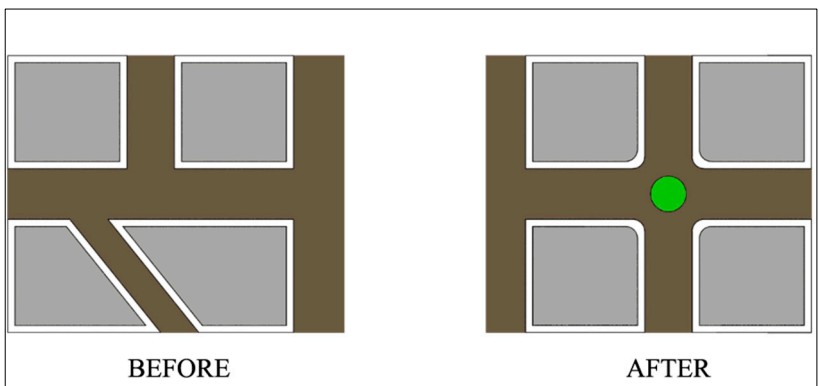

**Type 4.** Changes relating to transport

**Figure 4.** Plan changes under Type 1, Type 2, Type 3 and Type 4 (produced by the authors, 2021).

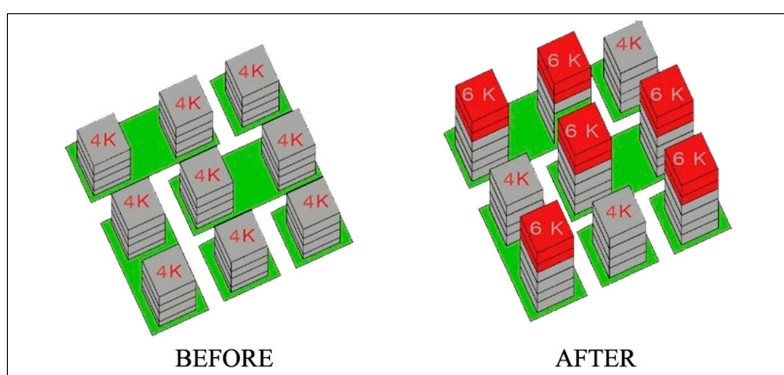

**Type 5.** Changes relating to an increase in the floor area ratio (FAR)

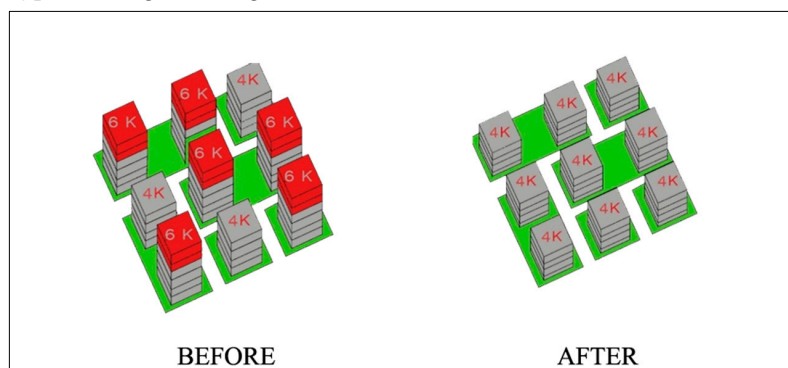

**Type 6.** Changes relating to a decrease in the floor area ratio (FAR)

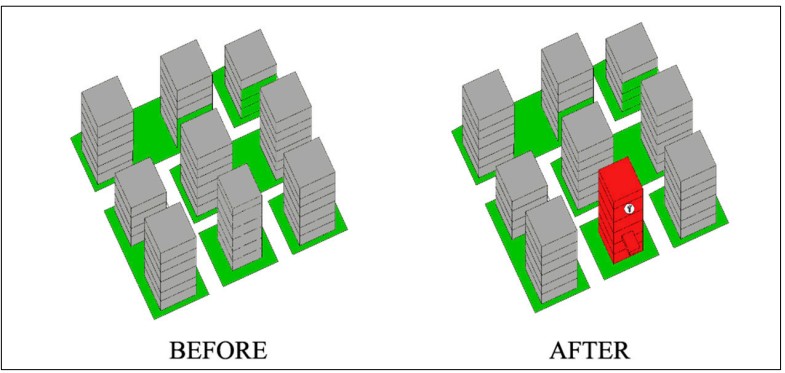

**Type 7.** Changes relating to usage area' (causing density, transport and environmental burden)

**Figure 5.** *Cont.*

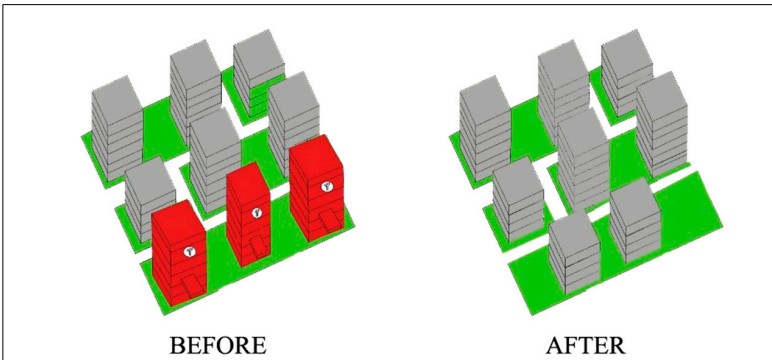

**Type 8.** Changes relating to usage area' (decreasing density, transport and environmental burden)

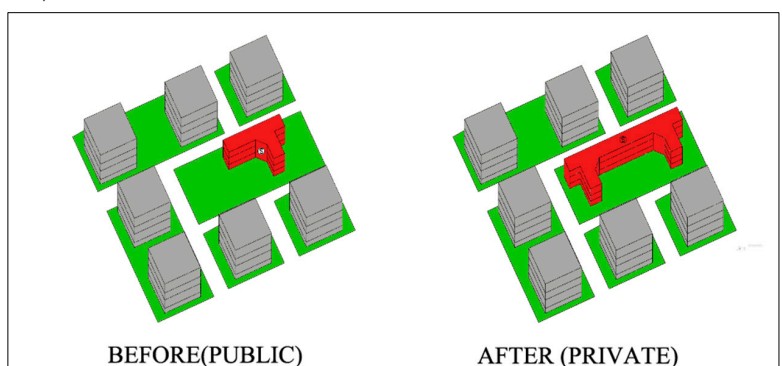

**Type 9.** Changes relating to private social technical infrastructure area

**Figure 5.** Plan changes under Type 5, Type 6, Type 7, Type 8 and Type 9.

### 4.3.2. The Second Stage: The Value Capture Content of Each Group

A sample was chosen from the data set representing each group. These examples are important in terms of understanding the value capture content of each group (Table 3).

**Table 3.** Evaluations of the groups classified by planners from the perspective of land value.

| Interviews | Type 1 | Type 2 | Type 3 | Type 4 | Type 5 | Type 6 | Type 7 | Type 8 | Type 9 | Interviews | Type 1 | Type 2 | Type 3 | Type 4 | Type 5 | Type 6 | Type 7 | Type 8 | Type 9 |
|---|---|---|---|---|---|---|---|---|---|---|---|---|---|---|---|---|---|---|---|
| Interview-1 | 0 | +1 | −1 | +1 | +1 | −1 | +1 | 0 | +1 | Interview-24 | +1 | +1 | −1 | 0 | +1 | −1 | +1 | −1 | +1 |
| Interview-2 | 0 | +1 | −1 | 0 | +1 | −1 | +1 | −1 | +1 | Interview-25 | 0 | +1 | −1 | 0 | +1 | −1 | +1 | −1 | +1 |
| Interview-3 | 0 | +1 | −1 | 0 | +1 | −1 | +1 | −1 | +1 | Interview-26 | 0 | +1 | 0 | +1 | +1 | −1 | 0 | −1 | +1 |
| Interview-4 | 0 | +1 | −1 | +1 | +1 | −1 | +1 | −1 | +1 | Interview-27 | 0 | +1 | −1 | 0 | +1 | −1 | +1 | −1 | +1 |
| Interview-5 | 0 | +1 | −1 | +1 | +1 | −1 | +1 | −1 | +1 | Interview-28 | +1 | +1 | −1 | 0 | +1 | −1 | +1 | −1 | +1 |
| Interview-6 | 0 | +1 | −1 | 0 | +1 | −1 | +1 | −1 | +1 | Interview-29 | +1 | +1 | 0 | +1 | +1 | −1 | 0 | 0 | +1 |
| Interview-7 | 0 | +1 | −1 | +1 | +1 | −1 | +1 | −1 | +1 | Interview-30 | +1 | +1 | −1 | 0 | +1 | 0 | +1 | −1 | +1 |
| Interview-8 | 0 | +1 | −1 | 0 | +1 | −1 | +1 | −1 | +1 | Interview-31 | 0 | +1 | −1 | +1 | +1 | −1 | +1 | −1 | 0 |
| Interview-9 | 0 | +1 | −1 | +1 | +1 | −1 | +1 | −1 | +1 | Interview-32 | 0 | +1 | 0 | +1 | +1 | −1 | +1 | −1 | +1 |
| Interview-10 | +1 | +1 | 0 | 0 | +1 | −1 | +1 | −1 | +1 | Interview-33 | +1 | +1 | −1 | +1 | +1 | −1 | +1 | −1 | +1 |
| Interview-11 | 0 | 0 | −1 | 0 | +1 | −1 | +1 | −1 | +1 | Interview-34 | 0 | +1 | −1 | 0 | +1 | −1 | +1 | −1 | +1 |
| Interview-12 | 0 | 0 | −1 | 0 | +1 | −1 | +1 | −1 | +1 | Interview-35 | +1 | +1 | −1 | +1 | +1 | −1 | +1 | −1 | +1 |
| Interview-13 | +1 | +1 | −1 | 0 | +1 | −1 | +1 | 0 | +1 | Interview-36 | +1 | +1 | 0 | +1 | +1 | −1 | +1 | −1 | +1 |
| Interview-14 | 0 | +1 | 0 | 0 | +1 | −1 | +1 | −1 | 0 | Interview-37 | 0 | +1 | 0 | 0 | +1 | −1 | +1 | −1 | +1 |
| Interview-15 | +1 | +1 | −1 | +1 | +1 | −1 | +1 | −1 | +1 | Interview-38 | +1 | +1 | 0 | +1 | +1 | −1 | +1 | −1 | +1 |
| Interview-16 | 0 | +1 | −1 | 0 | +1 | −1 | +1 | −1 | +1 | Interview-39 | +1 | +1 | −1 | 0 | +1 | −1 | +1 | 1 | +1 |

**Table 3.** *Cont.*

| Interviews | Type 1 | Type 2 | Type 3 | Type 4 | Type 5 | Type 6 | Type 7 | Type 8 | Type 9 | Interviews | Type 1 | Type 2 | Type 3 | Type 4 | Type 5 | Type 6 | Type 7 | Type 8 | Type 9 |
|---|---|---|---|---|---|---|---|---|---|---|---|---|---|---|---|---|---|---|---|
| Interview-17 | +1 | +1 | −1 | +1 | +1 | −1 | +1 | −1 | +1 | Interview-40 | 0 | +1 | 0 | +1 | +1 | −1 | +1 | −1 | 0 |
| Interview-18 | 0 | +1 | 0 | 0 | +1 | −1 | +1 | 0 | 0 | Interview-41 | 0 | +1 | 0 | 0 | +1 | −1 | +1 | −1 | 0 |
| Interview-19 | 0 | +1 | −1 | 0 | +1 | −1 | 0 | 0 | +1 | Interview-42 | +1 | +1 | −1 | 0 | +1 | −1 | +1 | −1 | +1 |
| Interview-20 | +1 | +1 | −1 | +1 | +1 | −1 | +1 | −1 | +1 | Interview-43 | +1 | +1 | −1 | +1 | +1 | −1 | +1 | −1 | +1 |
| Interview-21 | 0 | +1 | −1 | +1 | +1 | 0 | +1 | −1 | +1 | Interview-44 | 0 | +1 | −1 | +1 | +1 | −1 | +1 | −1 | +1 |
| Interview-22 | 0 | +1 | −1 | +1 | +1 | −1 | +1 | −1 | 0 | Interview-45 | +1 | +1 | 0 | 0 | +1 | 0 | +1 | 0 | +1 |
| Interview-23 | +1 | +1 | −1 | 0 | +1 | −1 | +1 | −1 | 0 | Interview-46 | +1 | +1 | −1 | +1 | +1 | −1 | +1 | −1 | +1 |
| Avarage | 0.26 | 0.91 | −0.9 | 0.43 | +1 | −0.96 | 0.96 | −0.83 | 0.83 | | 0.57 | +1 | −0.6 | 0.52 | +1 | −0.9 | 0.91 | −0.83 | 0.87 |

Note: Sample is 46 planners. "Value-increasing" (+1), "Value-decreasing" (−1) or "Ineffective" (0).

As seen in Table 3, according to the interviewees, not all of the plan changes may give rise to an increase in value. For instance, while, according to the interviewees, plan changes within the scope of Types 1 and 4 do not have a noteworthy impact on land value, those within the scope of Types 3, 6 and 8 have a decreasing effect on land value. The reason is that these plan changes reduce the development rights on a parcel, or they altogether eliminate such development rights and include the parcel into the facility areas.

On the other hand, according to the results obtained from the interviews, plan changes within the scope of Type 2 "Changes from a Social and Technical Infrastructure Area to a Usage Area", Type 5 "Changes Relating to an Increase in the Floor Area Ratio (FAR)", Type 7 "Changes Relating to Usage Area (Causing Density, Transport and Environmental Burden)" and Type 9 "Changes Relating to Private Social Technical Infrastructure Area" are those that would cause an increase in land value. Additionally, plan changes within the scope of Types 1 and 4, while not to the extent of the plan changes made in line with private sector demands, constitute another group of plan changes that increase land value. However, while these other plan changes that increase land value emerge as a result of the increasing development rights, an infrastructure-based value increase occurs in the plan changes within the scope of Types 1 and 4. Table 4 demonstrates the interviewees' remarks relating to the value capture capacities of the plan changes.

**Table 4.** The value capture capacities of the plan change groups according to evaluations of the interviewees.

| Interviews | Type 1 | Type 2 | Type 3 | Type 4 | Type 5 | Type 6 | Type 7 | Type 8 | Type 9 | Interviews | Type 1 | Type 2 | Type 3 | Type 4 | Type 5 | Type 6 | Type 7 | Type 8 | Type 9 |
|---|---|---|---|---|---|---|---|---|---|---|---|---|---|---|---|---|---|---|---|
| Interview-1 | 1 | 5 | 0 | 4 | 5 | 0 | 5 | 0 | 5 | Interview-24 | 2 | 5 | 1 | 2 | 5 | 0 | 5 | 0 | 5 |
| Interview-2 | 0 | 5 | 0 | 1 | 5 | 0 | 5 | 1 | 5 | Interview-25 | 1 | 5 | 0 | 1 | 5 | 1 | 4 | 0 | 4 |
| Interview-3 | 2 | 4 | 0 | 1 | 5 | 0 | 5 | 0 | 5 | Interview-26 | 1 | 5 | 0 | 2 | 5 | 0 | 5 | 0 | 5 |
| Interview-4 | 1 | 5 | 0 | 3 | 5 | 0 | 5 | 0 | 5 | Interview-27 | 2 | 5 | 1 | 3 | 5 | 0 | 5 | 0 | 5 |
| Interview-5 | 1 | 5 | 1 | 3 | 5 | 0 | 5 | 0 | 5 | Interview-28 | 1 | 5 | 0 | 1 | 5 | 1 | 5 | 1 | 5 |
| Interview-6 | 2 | 4 | 0 | 2 | 5 | 0 | 5 | 0 | 5 | Interview-29 | 3 | 5 | 0 | 4 | 5 | 0 | 4 | 0 | 3 |
| Interview-7 | 4 | 5 | 0 | 4 | 5 | 0 | 4 | 1 | 5 | Interview-30 | 1 | 5 | 0 | 1 | 5 | 0 | 5 | 0 | 5 |
| Interview-8 | 1 | 5 | 0 | 1 | 5 | 0 | 5 | 0 | 4 | Interview-31 | 0 | 5 | 0 | 3 | 5 | 0 | 5 | 0 | 5 |
| Interview-9 | 3 | 5 | 0 | 4 | 5 | 0 | 5 | 0 | 5 | Interview-32 | 1 | 5 | 0 | 4 | 5 | 0 | 5 | 0 | 5 |
| Interview-10 | 1 | 5 | 0 | 2 | 5 | 0 | 5 | 0 | 5 | Interview-33 | 3 | 5 | 0 | 3 | 5 | 0 | 5 | 0 | 5 |
| Interview-11 | 1 | 5 | 0 | 1 | 5 | 0 | 5 | 0 | 5 | Interview-34 | 1 | 5 | 0 | 1 | 5 | 0 | 5 | 0 | 5 |
| Interview-12 | 3 | 5 | 0 | 4 | 5 | 0 | 5 | 1 | 5 | Interview-35 | 1 | 5 | 2 | 3 | 5 | 1 | 5 | 0 | 4 |

**Table 4.** *Cont.*

| Interviews | Type 1 | Type 2 | Type 3 | Type 4 | Type 5 | Type 6 | Type 7 | Type 8 | Type 9 | Interviews | Type 1 | Type 2 | Type 3 | Type 4 | Type 5 | Type 6 | Type 7 | Type 8 | Type 9 |
|---|---|---|---|---|---|---|---|---|---|---|---|---|---|---|---|---|---|---|---|
| Interview-13 | 1 | 5 | 0 | 2 | 5 | 0 | 3 | 0 | 5 | Interview-36 | 1 | 5 | 0 | 4 | 5 | 0 | 5 | 0 | 5 |
| Interview-14 | 4 | 4 | 0 | 3 | 5 | 0 | 5 | 0 | 4 | Interview-37 | 2 | 5 | 0 | 1 | 5 | 0 | 5 | 0 | 5 |
| Interview-15 | 2 | 5 | 0 | 1 | 5 | 0 | 5 | 0 | 5 | Interview-38 | 1 | 5 | 1 | 3 | 5 | 1 | 5 | 0 | 5 |
| Interview-16 | 0 | 5 | 0 | 2 | 5 | 0 | 5 | 0 | 5 | Interview-39 | 1 | 5 | 0 | 1 | 5 | 0 | 4 | 0 | 5 |
| Interview-17 | 4 | 5 | 0 | 2 | 5 | 0 | 5 | 0 | 3 | Interview-40 | 3 | 5 | 0 | 4 | 5 | 1 | 4 | 1 | 4 |
| Interview-18 | 1 | 5 | 1 | 3 | 5 | 1 | 4 | 1 | 5 | Interview-41 | 2 | 5 | 1 | 1 | 5 | 0 | 5 | 0 | 5 |
| Interview-19 | 0 | 5 | 0 | 3 | 5 | 0 | 5 | 0 | 4 | Interview-42 | 1 | 5 | 0 | 1 | 5 | 1 | 5 | 0 | 4 |
| Interview-20 | 3 | 5 | 0 | 4 | 5 | 0 | 5 | 0 | 5 | Interview-43 | 1 | 5 | 0 | 4 | 5 | 0 | 5 | 0 | 5 |
| Interview-21 | 3 | 5 | 0 | 2 | 5 | 1 | 3 | 0 | 4 | Interview-44 | 3 | 5 | 2 | 4 | 5 | 0 | 5 | 1 | 5 |
| Interview-22 | 4 | 5 | 1 | 3 | 5 | 0 | 5 | 1 | 5 | Interview-45 | 1 | 5 | 0 | 1 | 5 | 1 | 5 | 0 | 4 |
| Interview-23 | 1 | 5 | 0 | 2 | 5 | 0 | 5 | 0 | 4 | Interview-46 | 1 | 5 | 0 | 3 | 5 | 0 | 5 | 1 | 5 |
| Avarage | 1.09 | 4.87 | 0.13 | 1.87 | 5 | 0.09 | 4.87 | 0.13 | 4.78 | | 1.48 | 5 | 0.35 | 1.87 | 5 | 0.3 | 4.91 | 0.17 | 4.7 |

Note: Sample is 46 planners. "No capacity" (0), "Very low capacity" (1), "Low capacity" (2), "Medium capacity" (3), "High capacity" (4) and "Very high capacity" (5).

The value capture capacities of the plan change groups were evaluated as "no capacity (0)", "very low capacity (1)", "low capacity (2)", "medium capacity (3)", "high capacity (4)" and "very high capacity (5)". As seen in Table 4, not all of the plan changes have similar value capture capacities. While the interviewees evaluated the value capture capacities of the plan changes within the scope of Types 3, 6 and 8 as "no capacity", they considered the value capture capacities of the plan changes under Types 1 and 4 as "very low". On the other hand, the interviewees evaluated the value capture capacities of the plan changes under Types 2, 5, 7 and 9 as "very high capacity". Among these changes, those with the greatest number are the plan changes under Type 2 (4031) and Type 5 (2877). Figure 6 provides a density analysis of the distribution of the plan changes, which increase land value and have a value capture capacity, throughout the Istanbul Metropolitan Area.

As seen in Figure 5, although the plan changes under Types 2, 5, 7 and 9 may have spread across the Istanbul Metropolitan Area, they are mostly concentrated in primary central districts such as Şişli, Beyoğlu, Üsküdar and Kadıköy, and secondary central districts such as Bağcılar, Zeytinburnu, Kağıthane Eyüpsultan, Gaziosmanpaşa, Ataşehir and Ümraniye. These districts are those whose development is relatively recent, as their development has gained momentum especially after 1990, and which are considerably dynamic in terms of population development. At the same time, these districts feature a high settlement pressure and high accessibility. This may also be interpreted as the private sector's desire to take a smaller amount of risks. Likewise, although the accessibility value of central and secondary-central districts is the highest, these districts, having been settlement areas for many years, are also the districts that have the lowest ratio of facility areas and green areas per capita [80]. For instance, the Bağcılar district is Istanbul's poorest district in terms of urban infrastructure and green areas, but it is also the district with the largest number of Types 2, 5, 7 and 9 plan changes (618), which have the effect of decreasing the amount of social and technical infrastructure. This makes the already-problematic issue of the amount of urban infrastructure and green area per capita even further problematic. Since the plan changes in this scope are performed more in line with the market's interests rather than for public benefit, they cause a decrease in urban resilience. Plan changes made for the benefit of the private sector are mostly of a nature that increases building and population density. They mostly focus on urban outdoor areas and urban infrastructure areas [50]. This causes an increase in the density of buildings and population in the urban space, as well as negative externalities such as the inadequacy of infrastructure areas and the impairment of the urban infrastructure-population balance.

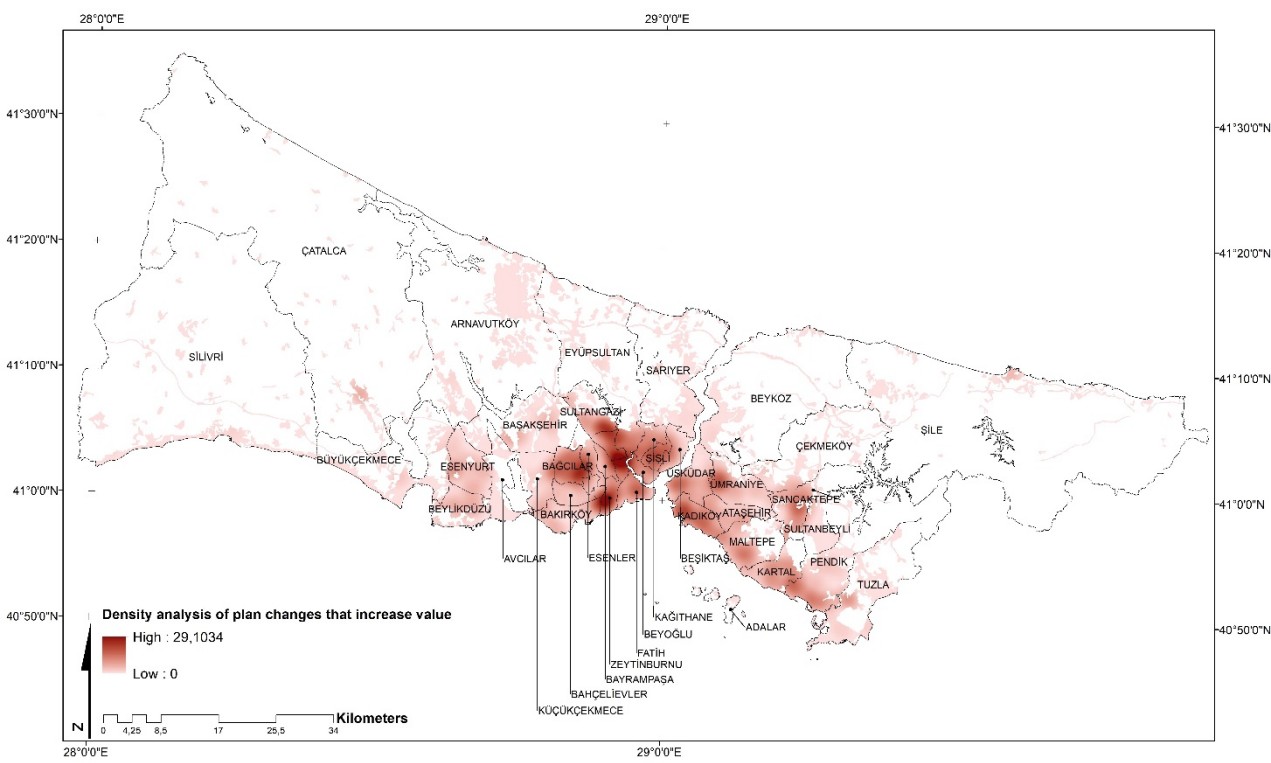

**Figure 6.** Density analysis of plan changes that increase value (produced by the authors, 2021).

The analyses performed under the study found that certain plan change groups have a decreasing effect on land value. The plan changes of this nature are Type 3 (changes from a usage area into a social and technical infrastructure area), Type 6 (changes relating to a decrease in floor area ratio) and Type 8 (changes relating to the usage area—decreasing density, transport and environmental burden). Through the plan changes under Type 3 from among these plan change groups, areas allocated to urban use such as housing and commerce are included in the urban infrastructure areas such as those for education, healthcare and green areas. The landowner is paid compensation in exchange. Therefore, it is not necessary to pay separate compensation for the value increase caused by plan changes under Type 3. However, expropriation of these areas after their inclusion into the development programmes takes time. The landowners are not able to make any construction on their land during such a period. These areas are sold with difficulty and fetch low prices. Due to this aspect, plan decisions impose a restriction on land. The compensation of this restriction by local governments would be fair and equitable. On the other hand, plan changes under Types 6 and 8 involve a value decrease stemming from the restriction of development rights. Since expropriation does not occur in the plan changes within this scope, no payment is made to the landowner. While this situation serves the public benefit in general, the interests of parcel owners are thereby adversely affected. The value decrease occurring as a result of the plan changes under Types 6 and 8 must be compensated by the planning units in order to establish a balance between public benefit and individual loss. Figure 7 provides a density analysis of the distribution, throughout the Istanbul Metropolitan Area, of the plan changes that decrease land value and require the payment of compensation.

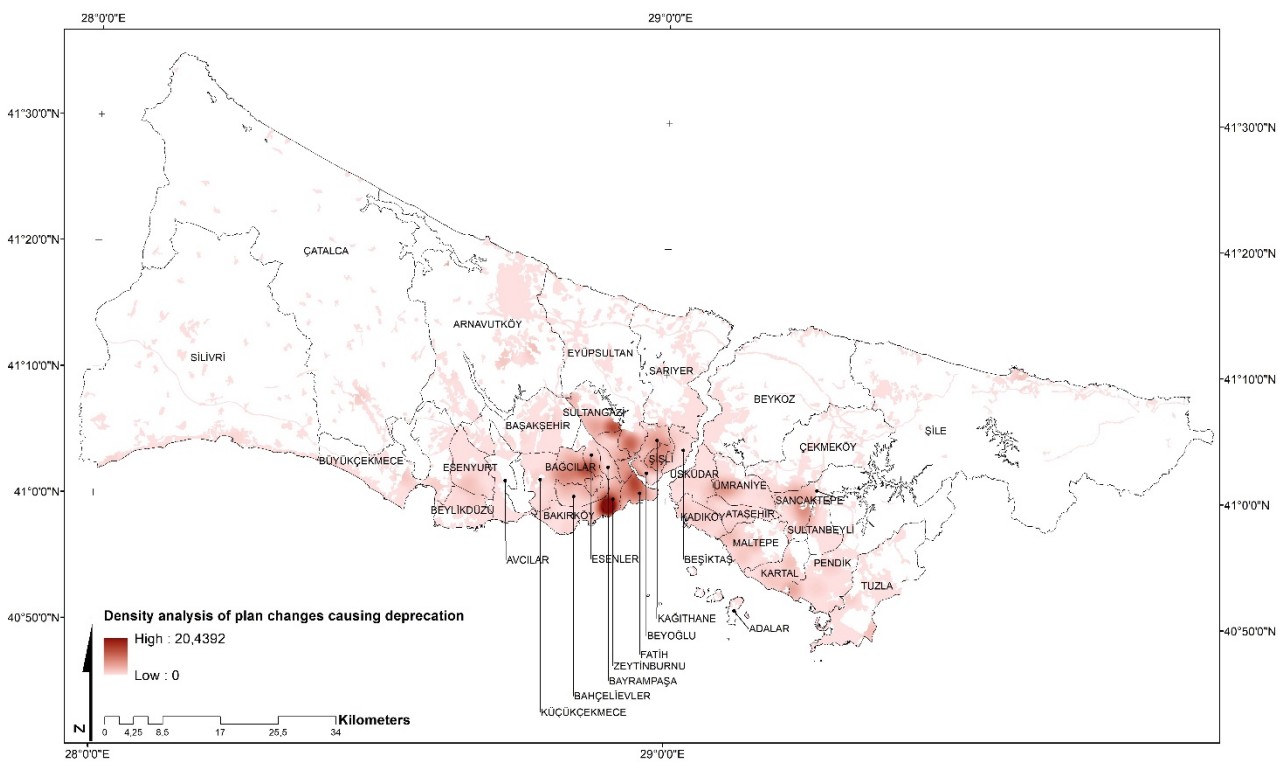

**Figure 7.** Density analysis of plan changes causing depreciation (produced by the authors, 2021).

As seen in Figure 7, the plan changes in the scope of Types 3, 6 and 8 are more frequently observed in the Fatih (historic peninsula), Eyüpsultan and Zeytinburnu districts. The districts in question are also urban conservation areas. Plan changes under Types 6 and 8 that restrict development rights are mostly performed in these areas. While the plan changes under Types 6 and 8 benefit the city as a whole by decreasing the density of buildings and population, they also cause owners of parcels that are the subject of changes to suffer grievances by restricting their construction rights. On the other hand, the plan changes under Type 3, which is another group of plan changes that cause a decrease in value, are concentrated in the Bağcılar district. It is among the districts with the smallest amount of urban infrastructure and green areas per capita. The fact that plan changes within the scope of Type 3, which aim at converting the usage areas that are private ownership, such as for housing and commerce, into urban infrastructure areas such as those for education, healthcare and green space, are most intensely observed in the Bağcılar district, there is an increase in urban infrastructure standards in this area. However, the plan changes in this scope altogether eliminate the construction rights of landowners in the areas in which the changes are performed. To counter this, expropriation or land readjustment may be carried out when such areas are taken over for public purposes. In both cases, it is possible to compensate the landowner with respect to the situation brought about by the plan decision. It becomes necessary to give the landowner compensation in the case of expropriation. Although compensations are made based on current floor area ratios, expropriation procedures cannot be carried out immediately after the plan decisions, which also imposes a restriction on the construction rights of the parcel, until it is expropriated. The sale of property in these areas either cannot be performed by landowners or takes place at much lower prices compared to the market prices. Another practice is the implementation of the land readjustment. The balance between betterment and compensation can be achieved through the implementation of land readjustment. However, land readjustment in Turkey involves certain restrictions [81]. Therefore, it is possible that local governments may also not make effective use of the land and plot arrangement. In other words, the delay in expropriation procedures or the failure to carry

out land readjustment cause grievance to landowners in the plan changes under Type 3. Plan changes under Types 6 and 8, on the other hand, offer no compensation against a drop of value in property belonging to landowners.

## 5. Discussion

In this study, plan changes made in Istanbul (Turkey) between 2009 and 2018 were considered, and the value capture capacity of plan changes was discussed. The relationship between partial interventions and value capture is conceptualised differently in the international literature. This study examines the value capture in light of the spatial impacts of partial interventions, and the effectiveness of the tools developed to eliminate these impacts.

Identifying the contents of a plan change and determining whether or not such change would cause a value increase are primary issues. According to the findings of the study, the plan changes under Types 2, 5, 7 and 9 caused an increase in value. For this reason, it is noteworthy that a value capture policy was developed only to address plan changes that remain in this scope. On the other hand, plan changes in the scope of Type 3 (changes from a usage area into a social and technical infrastructure area), Type 6 (changes relating to a decrease in floor area ratio) and Type 8 (changes relating to the usage area—decreasing density, transport and environmental burden) are those that cause a decrease in value. In the plan changes in the scope of Type 3, parcels are needed either be expropriated, and expropriation payments will be made to landowners, or they will be the subject of land readjustment. Therefore, no extra payment is required to compensate for value decrease. However, due to the fact that the expropriation of these areas takes a long time in practice or land readjustment cannot be implemented, landowners are not able to do any construction on the parcel during such process. On the other hand, the sale of these parcels is possible only at prices that are below their market values. In other words, although an expropriation payment will be made following a plan decision, there exists a restriction on the land until expropriation is carried out. On the other hand, up to 45% of these areas can be allocated to public use without any payment as a result of land readjustment. It is clear that no compensation can be provided if no land readjustment can be made. Paying compensation to the property owner in return for such restrictions is an equitable approach. Policies such as paying a planning price, as in Poland, may be developed for cases where the ownership right on the land is fully or partially restricted as a result of plan decisions. Contrarily, since no expropriation price or compensation is paid in the event of plan changes under Types 6 and 8, the payment of the decrease in value to landowners would be an equitable approach.

In metropolitan areas, especially in Istanbul, the existing urban infrastructure proves inadequate to meet the needs of the increasing population. Infrastructure investments cost large amounts of money. Urban infrastructure areas across Istanbul are far below the standards prescribed in the regulation. Therefore, identifying the value increase occurring as a result of plan changes under Types 2, 5, 7 and 9, and transferring part of the value increase calculated on plot and parcel basis to the account of district municipalities, and another part to the account of metropolitan municipalities, may be considered as an option. Since the adverse externalities of plan changes mostly impact their close vicinity, transferring a larger part of the collected amount at a local level—district municipality—would be a more correct approach. Also, part of this income could also be used in compensation for the value decrease resulting from the plan change.

## 6. Conclusions

Plan changes offer an important potential for the capture of value increase to the public. Development rights are generally increased through plan changes, which, in turn, give rise to an increase in land value. However, the value increase that occurs in land through a plan change, which is a public decision, is not always captured to the public, and is generally shared between landowners and investors. This situation makes plan changes

more attractive in the eyes of investors. On the other hand, governments of developing countries avoid making policies for capturing the value arising as a result of plan changes to the public for fear of losing voters. As a result, plan changes become a tool that serves the enrichment of landowners and investors.

There is an international trend with respect to value capture that involves the more effective use of indirect tools rather than employing direct value tools [2,3,9]. This arises from governments' belief that indirect value capture tools are less conspicuous to voters and are more flexible and practical compared to direct tools. Although a recent tendency may be observed in Turkey towards an increase in the use of indirect tools, the regulatory framework relating to the recovery of value increase caused by plan changes is prepared as a mixture of direct and indirect value capture.

On the other hand, the international literature recognises that partial interventions mostly increase land value. However, as can be seen in the analyses conducted on the Istanbul Metropolitan Area as the area of the study, there are also partial interventions that do not affect land value, and in fact decrease it. Nevertheless, both the literature and legal sources mostly fail to take notice of the cases where partial interventions cause value decrease.

The Turkish planning system that has offered no rules on ensuring the return of the increase in value to the public until February 2020 has increased the pressure towards change on plans. Furthermore, the legal framework rests on the assumption that all plan changes would cause a value increase, regardless of their contents. However, as can be seen in this study, there are also plan changes that cause no value increase, but rather value decrease. As can be seen in the findings of this study, although there has been a great number of plan changes, most of these changes were made in favour of the private sector. The value increase that occurred in plan changes carried out by the private sector to maximise profit was shared between the investor and the landowners, and the return of this increase to the public was not achieved. Due to political concerns, governments mostly ignored the issue relating to the taxation of the income generated as a result of plan changes, and even allowed plan changes that would encourage investors.

It may be suggested that a special fund to collect the values arising from plan changes be created within municipalities. Part of the monies collected in district municipalities' accounts may be used in expropriating the areas allocated to functions such as green areas, roads and municipal service areas, while another part may be transferred to the urban transformation accounts, and yet another part to infrastructure costs to be spent for Istanbul. Additionally, this value increase that is achieved may be used in compensating for areas the value of which has decreased due to plan changes. This is because expropriation, urban transformation and infrastructure expenditures are among the most important expense items of municipalities. This way, in addition to creating resources for expropriating parcels remaining within the social and technical infrastructure areas in the plans, increasing the speed of urban transformation activities, and improving the infrastructure, it may also be possible to make sure that the social and technical infrastructure areas shown in the plans are taken over for public use more quickly.

**Author Contributions:** Conceptualization, N.K. and S.S.T.; methodology, N.K. and S.S.T.; validation, N.K. and S.S.T.; formal analysis, N.K. and S.S.T.; investigation N.K. and S.S.T.; resources N.K. and S.S.T.; data curation, N.K. and S.S.T.; writing—original draft preparation, N.K. and S.S.T.; writing—review and editing, N.K. and S.S.T.; visualization, N.K.; supervision, S.S.T.; project administration N.K. and S.S.T.; funding acquisition, N.K. and S.S.T. All authors have read and agreed to the published version of the manuscript.

**Funding:** This research received no external funding.

**Institutional Review Board Statement:** Not applicable.

**Informed Consent Statement:** Not applicable.

**Data Availability Statement:** Istanbul metropolitan municipality council decisions used in the article can be accessed at https://ibb.istanbul/Council/Decision, accessed on 3 December 2021.

**Conflicts of Interest:** The authors declare no conflict of interest.

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
