# Peer review of "Examination of Local Plan Changes from a Value Capture Perspective: Istanbul Case"

_sustainability, doi:10.3390/su14010329_

Round 1
Reviewer 1 Report
1、The abstract is not very clear, the main purpose of writing the abstract is to describe what kind of result we got from our research, and what is the principal significance of this result,so I suggest rewriting the abstract.
2、The study area is not described very well. The description of the study area has to rewrite, also have to add the map of the study area.
3、There is no any longitude and latitude in all of pictures, so I suggest adding the longitude and latitude all of the pictures.
4、The introduction part should add some preliminary related research at home and abroad.
5、the result section is too long, but there is no discussion part in the paper, so the discussion section should be written separately
6、I also suggest rewriting the conclusion part.
Reviewer 2 Report
I think that the topic covered by the paper is very interesting and can have some feedback both theoretically and practically. I suggest only to check the images inserted because they are not displayed correctly.
I enclose the pdf of the paper with some notes.

Round 2
Reviewer 1 Report
This paper has been revised according to my suggestions and can be published present form